# Reconstructing three decades of total international trawling effort in the North Sea

Elena Couce[1], Michaela Schratzberger[1,2], Georg H. Engelhard[1,2]

[1]Centre for Environment, Fisheries & Aquaculture Science (Cefas), Pakefield Road, Lowestoft NR33 0HT, UK
[2]Collaborative Centre for Sustainable Use of the Seas (CCSUS), University of East Anglia, Norwich NR4 7TJ, UK

*Correspondence to*: Elena Couce (elena.couce@cefas.co.uk)

**Abstract.** Fishing – especially trawling – is one of the most ubiquitous anthropogenic pressures on marine ecosystems worldwide, yet very few long-term, spatially explicit datasets on trawling effort exist; this greatly hampers our understanding of the medium- to long-term impact of trawling. This important gap is addressed here for the North Sea, a highly productive
shelf sea which is also subject to many anthropogenic pressures. For a 31-year time span (1985–2015), we provide a gridded dataset of the spatial distribution of total international otter and beam trawling effort, with a resolution of 0.5° latitude by 1° longitude, over the North Sea. The dataset was largely reconstructed using compiled effort data from 7 fishing effort time-series, each covering shorter time spans and only some of the countries fishing the North Sea. For the years where effort data for particular countries was missing, the series was complemented using estimated (modelled) effort data. This new, long-
term and large-scale trawling dataset may serve the wider scientific community, as well as those involved with policy and management, as a valuable information source on fishing pressure in a Large Marine Ecosystem which is heavily impacted, but which simultaneously provides a wealth of ecosystem services to society. The dataset is available on the Cefas Data Hub at: https://doi.org/10.14466/CefasDataHub.61 (Couce et al., 2019).

## 1 Introduction

Coastal and shelf seas are of great value to human societies and, being more productive than open oceans, provide some 80% of the world's wild-capture fisheries (Watson et al., 2016). Yet the process of fishing that is required to obtain these benefits and services also exerts a major anthropogenic pressure on shelf seas worldwide – along with climate change, pollution, eutrophication and habitat loss (Hiddink et al., 2006; Jennings et al., 2016). Trawling is considered one of the more invasive fishing methods, as it does not only impact target fish populations (through removal of fish and size-selective harvesting) but
also has wider-ranging ecosystem effects, including on benthic organisms and habitats, and other non-target species (Hiddink et al., 2017; Jennings et al. 2001; Schratzberger et al. 2002). Unfortunately, there is a lack of available long-term, spatially explicit datasets on trawling effort, and this has hampered our understanding of the direct and indirect effects of trawling pressure on the marine environment (Collie et al., 2017; Jennings et al., 2001).

The North Sea is one of the world's most important shelf seas in terms of fisheries production – and has been so for centuries, 'feeding' some of the world's most densely populated areas (e.g., Capuzzo et al., 2018). Yet it is also subject to extensive anthropogenic pressures due to its geographical location in central Europe surrounded by seven countries, with concerns about pollution, habitat degradation, major ecosystem changes, and overfishing (Emeis et al., 2015; Kenny et al., 2018). Trawling, in particular, is seen as one of the most significant impacts on not only fish but also marine benthos in the North Sea (Kenny et al., 2018).

Two trawl fishing methods predominate in the North Sea, and generally in shelf seas worldwide: beam trawlers (defined as any vessels towing nets supported by a rigid beam, usually one lowered from each side of the vessel) and otter trawlers (defined as any vessels towing bottom-fishing nets held open by trawl doors; Engelhard, 2008; Jennings et al., 2001). Both fishing methods impact the seabed and marine life, although in subtly different ways: with beam trawlers especially catching flatfish and the gear having particularly close and invasive contact with the seabed and benthos; and otter trawlers especially catching roundfish and the gear having less close contact with the ground but often over a much larger area, and fish being caught over a higher 'vertical area' within the water column (Jennings et al., 2001).

The North Sea has been extensively studied in terms of ecology and oceanography, with historical datasets dating back to the late 19[th] or early 20[th] centuries (e.g., Engelhard et al., 2014; Morris et al., 2018; Rijnsdorp and Millner, 1996; Sguotti et al., 2016). This facilitates studies of long-term change which are rare in marine research. However, the availability of historical fishing effort data is very limited, because time spent fishing and location choices are often linked to commercial interests of the fishing industry. Reluctance to share such data has resulted in a scarcity of long-term spatially explicit temporal data on fishing pressure. This paper aims at addressing this gap by presenting a 31-year long, spatially detailed dataset of total international trawling effort for the North Sea, distinguishing between otter and beam trawlers. There have been various previous attempts at putting together spatio-temporal datasets on trawling effort for the North Sea region, which have provided partial snapshots of the fishing in what is one of the most intensively exploited regions of the world (Callaway et al., 2002; Jennings et al., 1999). Unfortunately, while such evidence is available for distinct periods (e.g., see STECF, 2017 for the more recent period, and Jennings et al., 1999 for the early 1990s), it is not available for longer, multidecadal time-spans. Here we compile existing datasets, and "fill in the gaps" by estimating likely country-level fishing effort in periods for which 'nominal' data was lacking, in order to reconstruct as complete a picture as possible for the period from 1985 to 2015. We envisage that the trawling effort data reconstructed here will be of great use for researchers who seek to understand the impacts of commercial fisheries on marine organisms, making use of the plethora of other historical datasets available in this region.

## 2. Methods

For the 31-year period from 1985 to 2015, we collated or estimated data on total (demersal) otter and beam trawling effort per year for the North Sea, defined as ICES (International Council for the Exploration of the Sea) Sub-area IV. Specifically,

the data was spatially separated to the level of ICES statistical rectangles (1° latitude by 0.5° longitude). We did so for the demersal trawling effort by vessels landing in Belgium, Denmark, England, France, Germany, the Netherlands, Norway, Scotland and Sweden (in the case of Sweden otter trawling only, since its contribution to beam trawling effort is absent or negligible; see STECF, 2017). These countries are the most significant contributors to trawling effort in the North Sea region, together comprising >99% of the total effort (García-Carreras et al., 2015; Greenstreet et al., 2007; STECF, 2017). The effort was quantified as number of hours fishing in a year per ICES rectangle, recorded separately for beam and otter trawling (Couce et al., 2019).

## 2.1 Compilation of existing datasets on trawling effort

Seven datasets on trawling effort were included, covering different intervals within our 1985-2015 study period (see Fig. 1a for an overview), with only one of these, trawling by vessels landing into England, covering the full time-span examined. Each of these datasets included either one or multiple countries, and in the latter case, two datasets provided only the aggregated total for multiple countries combined and not for each country separately (but disaggregated by rectangle). In the following paragraphs, we briefly describe all datasets used.

For the earliest period until 1995, data was collated from Jennings et al. (1999) who assembled two different trawling pressure datasets from the North Sea, differing in time-span covered and countries included. The first of these (here referred to as 'Jennings et al. dataset 1,' see Fig. 1a) compiled effort data for 1977-1995 by English, German, Norwegian, Scottish and Welsh vessels. The second of these (here referred to as 'Jennings et al. dataset 2,' see Fig. 1a) covered a shorter time-span (1990-1995) but included effort by Danish and Dutch vessels in addition to those in the first dataset. For both of these datasets, only the data aggregated over all countries included was available, with no information on separate countries' contributions to the total (as had been agreed *a priori* by the different countries' institutions participating in the study). More details on the data and its sources can be found in Jennings et al. (1999).

The MAFCONS project ('Managing Fisheries to Conserve Groundfish and Benthic Invertebrate Species Diversity', www.mafcons.org) assembled data on demersal trawling and seining effort in the North Sea for the period 1997-2002 for Dutch, German, Norwegian, and UK vessels (Greenstreet et al., 2007). As in Jennings et al. (1999), data was aggregated as hours fishing by ICES rectangle. For Dutch and Scottish vessels this had to be estimated, since the data was provided as 'days absent from port' rather than number of hours fishing (for the method followed, see Greenstreet et al., 2007, 2009). Unlike for Jennings et al. (1999), total effort was broken down into individual country contributions. Data for the effort of beam trawling for the German fleet in the MAFCONS dataset does not include shrimp trawls. This, however, is included in the other compiled datasets and represents a significant contribution to the total beam trawling pressure in the North Sea. Therefore, for consistency we did not to use the MAFCONS beam trawling data for Germany and instead estimated it for this period. Although the MAFCONS dataset also included seining effort, only data on demersal otter trawl and beam trawl effort was considered for the present study (referred to as 'MAFCONS dataset' in Fig. 1a).

From 2002 onwards, compilation of data on trawling effort by European Union countries in the North Sea and adjacent waters has been carried out by the Scientific, Technical and Economic Committee for Fisheries (STECF) of the European Commission. Member States are required to submit fishing effort data to STECF, in response to the Data Collection Framework (DCF) Fishing Effort Regimes Data Call in 2013 (Martinsohn, 2014). STECF spatial effort data is available as annual fishing hours per ICES rectangle, for different gear types and vessel size classes. For the present study, annual data for Belgian, Danish, Dutch, English, French, German, Scottish, and Swedish vessels over 15 meters, was downloaded on February 23rd, 2017 from https://stecf.jrc.ec.europa.eu/dd/effort/graphs-quarter. For two countries – Belgium and France – effort data was available from 2000 onwards, and for the other countries from 2002 onwards. The classification of gear types in STECF data follows definitions outlined in Annex I of Regulation 1342/2008 (Council of the European Union, 2008). For the present study, gears defined by STECF as 'BEAM', 'BT1' and 'BT2' were included in our 'Beam trawling' category, whereas 'OTTER', 'TR1', 'TR2', 'TR3' were included as 'Otter trawling' (in line with Engelhard et al., 2015; García-Carreras et al., 2015).

Three additional effort datasets were also collated to complement our study (see Fig. 1a). For the period 1985-2012, data on otter trawling effort by vessels landing into Scotland was obtained from the Fisheries Management Database of Marine Scotland. For the full study period 1985–2015, data on beam and otter trawling effort by vessels landing into England and Wales was obtained from the Fisheries Activity Database of the Department for Environment, Food & Rural Affairs (Defra, UK). For the period 1987–2015, data on beam and otter trawling effort by vessels landing in Denmark (held at the Ministry of Food, Agriculture and Fisheries, Denmark) was kindly provided by Ole Ritzau Eigaard (pers. comm.; National Institute of Aquatic Resources [DTU-Aqua], Denmark).

## 2.2 Estimating missing data

In the years for which trawling effort data was lacking for certain countries, estimates of trawling effort by rectangle were reconstructed, based on two assumptions: (1) that the relative contributions of each country to the total trawling effort change slowly and gradually; and (2) that the spatial distribution of trawling over time changes slowly and gradually. Assumption (1) is tightly linked to the Common Fishery Policy's rule of 'relative stability,' whereby the quotas of all commercial fish stocks in the North Sea are allocated between countries according to a fixed allocation key, so that the distribution of fishing effort between countries will also be fairly constant; this is illustrated in Fig. 2 for a subset of all data included here (i.e., the STECF data). Assumption (2) partly relates to fishing vessels being based at particular ports, having traditional fishing grounds and fishing preferences, and having quotas associated with particular areas; these constraints imply that spatial distribution of fishing aggregated at fleet level will only change gradually from year to year (for examples of gradual change only in spatial distribution of fishing, see Engelhard, 2005; Greenstreet et al., 2007; Jennings et al., 1999). We acknowledge that over longer time-spans or under particular circumstances, major changes may occur. The outbreak of World War II in 1939, for example, brought fishing in the North Sea to a near standstill (Engelhard 2008). However, we are not aware of any such abrupt change taking place over our study period. Thus, in cases where a country was lacking effort

data for a particular year, effort was estimated based on the same country's average spatial distribution of effort over a close time period with available data, normalised so that the relative contribution of effort by the country compared to other countries was maintained.

The precise procedure followed to estimate the trawling effort for a period of $n$ consecutive years ("the missing period") for which a country ("the target country") lacks data was:

1) Estimating the spatial distribution of effort:

   a) Average of the spatial distribution of trawling effort for the target country in the $0.5n$ years before and $0.5n$ years after the missing period.

   b) When 1.a is not possible, use $n$ adjacent years (of if less than $n$ years are available, use them all).

2) Scaling the contribution:

   a) Using the longest time interval for which data is available for the target country, compute the average ratio between trawling by the target country and the aggregated trawling by as many other countries as possible with compiled data in the missing period, and normalise so that this ratio is maintained.

   b) When the missing period is covered by an aggregated dataset it is possible that no interval exists with data for both the target and all the countries in the aggregated dataset; in that case, use an interval with data for the target country and the majority of the other countries in the dataset, and estimate the contribution of the countries lacking data in that interval, following the procedure in step 2.a.

Table 1 summarises the missing periods that had to be estimated for all countries, and details how the estimation was carried out in each case (i.e., the periods and source data used when following steps 1 and 2 listed above). One exception to this procedure was the reconstruction of otter trawling effort for Scotland for 2013-2015. This data is actually included in the STECF dataset, but there was a significant mismatch between our Scotland dataset and that in STECF. Therefore, for 2013-2015 we normalised Scotland otter trawling effort in STECF by a correction factor which was the average of the annual total number of hours reported for Scotland in STECF versus our country dataset in 2003–2012.

In order to quantify the errors of the estimation of trawling pressure data in cases of missing values, for each country we calculated trawling effort by rectangle, using the same approach outlined above, but now for additional periods for which compiled data for that country was actually available. In that way, the differences between our estimates and the compiled data could be quantified. In each case, and when data allowed, the period that was estimated was chosen to be close in time and similar in duration to the 'real' missing periods. Additionally, the estimation rules listed above were adjusted so that, whenever possible, the procedure matched the one that had been followed for the estimation of the real missing periods. A median relative error between estimated and compiled data among all ICES rectangles and over the entire period was then computed, and applied to the estimated national trawling pressure data to produce a measurement of total absolute error of our estimations. Details on the periods that were estimated for each country and the data used in the estimation, together with the relevant median relative errors are listed in Table 2.

## 3. Results

We were able to estimate the total international beam trawl effort by rectangle in the North Sea for all years from 1985-2015 (Fig. 3) and, likewise, the total international otter trawling effort for the same period (Fig. 4). For the majority of years, but especially after 2000/2002, the reconstructed trawling effort by rectangle could be directly sourced from compiled data on

'nominal' trawling effort (see white sections of pie charts in Fig. 3 and 4) as opposed to estimated (black sections of pie charts). For some of the earlier years, there was less availability of compiled data and hence larger proportions of the reconstructed effort data had to be estimated. For beam trawl effort, >50% of reconstructed effort data was estimated in case of the years 1985–1989 and 1996, and for 1997–2002 the proportion was also close to 50%. For otter trawl effort, >50% of reconstructed effort data was estimated in case of the years 1985 and 1986 only. The greater scarcity of beam trawl effort

data in the 1980s was related to a lack of nominal effort data for the Netherlands, which is the country that generally predominates beam trawling in the North Sea. Since the proportion of estimated data is not dominating the reconstructed total for most years, the relative errors (Fig. 5 and 6) remain at very low levels for the majority of the North Sea during the study period. Exceptions to this are the earliest period until 1989 together with 1996 for beam trawling, and 1985-1986 for otter trawling, where a significant part of the study region reached relative error values around 0.5.

The spatial distribution of beam trawl effort in the North Sea (Fig. 3), based on our reconstructions, has generally remained fairly constant during 1985-2015, with a clear northwest–southeast gradient. Absolute levels of beam trawling were highest in the 1990s; since 2000, total beam trawl effort has declined and gradually become more concentrated in the shallower, eastern and south-eastern parts of the North Sea. Whilst our results indicate that in the 1980s–1990s there were appreciable levels of beam trawling off eastern and north-eastern Scotland, beam trawling in these areas has very much declined since

then.

No clear spatial gradient was evident for the distribution of otter trawl effort in the North Sea, which over the years 1985-2015 was generally spread more evenly throughout the region (Fig. 4). The overall levels of otter trawling have declined, especially since 2000. Within the North Sea, some localised areas stood out as undergoing greater otter trawl effort. These include areas off eastern Scotland (Moray Firth Ground, Wee Bankie); off north-east England (Farn Deeps, western Dogger

Bank); west of Denmark (Little Fisher Bank, Jutland Bank); and the southernmost rectangles within the North Sea (between the Thames estuary and Belgium). In many years, otter trawl effort was also high along the western slopes of the Norwegian Trench. The deeper parts of the Norwegian Trench received low otter trawl effort (Fig. 4). The shallower parts of the Southern Bight and German Bight, especially in recent years, received very little otter trawl effort (but highest levels of beam trawl effort; compare Fig. 3 and 4).

Although there have been changes in the total levels of otter trawling in the North Sea, there was evidence of fairly persistent spatial patterns; however, the relative contribution of trawling in the western North Sea off north-eastern England and Scotland was higher in the 1980s–1990s than in more recent years (Fig. 4).

**4. Data availability**

Reconstructed, nominal and estimated trawling effort data is available from the Cefas Data Hub (Couce et al.: Reconstruction of North Sea trawling effort 1985-2015, DOI: 10.14466/CefasDataHub.61, 2019).

The contents of the Cefas Data Hub website are provided as part of the Cefas role as a Defra agency under the Defra Open
Data Strategy.

Cefas requires users to make their own decisions regarding the accuracy, reliability, and applicability of information provided. The data provided by the Cefas Data Hub are believed by Cefas to be reliable for their original purposes and are accompanied by discovery metadata that provide a copy of the information available to Cefas scientists, describing the original purposes of data collection. It is the responsibility of the data user to take this information into account when reusing
data. Regardless of any quality control processes, Cefas does not accept any liability for the use of the data provided; use is at the users' own risk. Cefas does not give any warranty as to the quality or accuracy of the information or the medium on which it is provided or its suitability for any use. All implied conditions relating to the quality or suitability of the information and the medium and all liabilities arising from the supply of the information (including any liability arising from negligence) are excluded to the fullest extent permitted by law.

The use of data from the Cefas Data Hub requires that the correct and appropriate interpretation is solely the responsibility of the data users, that results, conclusions, and/or recommendations derived from the data do not imply endorsement from Cefas, that data sources must be acknowledged, preferably using a formal citation, that data users must respect all restrictions on the use of data such as for commercial purposes, and that data may only be redistributed, i.e. made available in other data collections or data portals, with the prior written consent of Cefas.

**5. Discussion**

This study represents the first reconstruction of total international trawling effort in the North Sea, spatially detailed by ICES rectangle, over a multi-decadal time span. The reconstructions were, as much as possible, based on compiled (nominal) effort data. Where such data was not available, efforts were made to 'fill in' any gaps by modelling effort estimations, and so provide a holistic picture of the total trawling pressure in the North Sea over the past 31 years. Earlier studies that have
attempted to compile international trawling effort in the North Sea have covered considerably shorter time spans (e.g. Jennings et al. 1999: period 1990–1995; Callaway et al., 2002: year 1998; Greenstreet et al. 2007: period 1997–2004; STECF 2017: 2002–2015; Engelhard et al. 2015: periods 1990–1995 and 2003–2012). Those studies moreover did not attempt to reconstruct data in cases where country-specific effort data was lacking for certain years (with the exception of Greenstreet et al. 2007).

Reconstruction of missing data may in some cases have led to erroneous estimations. In all cases, for the spatial distribution of the effort we attempted to use the most relevant country-specific data available, from a period close in time. Moreover, we have been transparent in keeping the compiled (nominal) and modelled (estimated) data separate, and when displaying totals

we have indicated the proportion of the data that was estimated (e.g., see black and white pie charts in Fig. 3 and 4). Likewise, Greenstreet et al. (2007) who attempted to reconstruct total international trawling in the North Sea for the 1997–2004 period, also had to model effort for some countries, which in their case was lacking for Belgium, Sweden, France and Denmark. They used a different approach to tackle this problem, based on a combination of landing, catch-per-unit-effort, and fleet size data. Encouragingly, in spite of the different approaches, their reconstructions of total international otter and beam trawl effort by rectangle are in broad agreement with those presented here (compare our Fig. 3 and 4 with pages 118-119 in Greenstreet et al. 2007).

We acknowledge discrepancies between our otter trawl effort data for England and Scotland (based on the national databases of England and Scotland) and the data collated by STECF since 2002 for these two countries. Effort data differ by roughly a factor of 2 in each case (our otter trawling effort data for Scotland is half that of STECF, and twice as much in the case of England). Although we cannot fully explain this discrepancy, we believe it relates to the conversion factor 24 assumed in the STECF compilation to convert from days-at-sea to number of hours fishing; but a considerable portion of time that fishing vessels are away from port is spent either steaming or spent handling the catch, with a variable portion spent in the actual fishing operations (see discussion, and supplementary materials, in Engelhard et al., 2015). This might, to some extent, have affected our estimations on spatial distribution of trawling. Given that Scotland has extensive otter trawl fisheries especially in its near waters, our maps might underestimate otter trawling effort in areas near Scottish coastlines (see Greenstreet et al., 1999 for a review and spatio-temporal patterns of Scottish trawl fisheries). No significant discrepancy was found between the Danish national dataset and the STECF-collated data over the period they overlap.

When the present study's effort distribution maps for specific years are compared with earlier studies, some differences may be noted. For example, for the period 1990–1995 our trawling reconstructions, compared to Jennings et al. (1999), indicate higher levels of otter trawl effort in the area north-west of Denmark. This difference relates to the inclusion of Danish otter trawling in our study, which was likely omitted in Jennings et al. (1999), and suggests that the benthic environment in this particular area was subjected to greater anthropogenic pressure than previously assumed. For the year 1998, however, a very close spatial match of our trawling reconstruction was noted, compared with that collated by Callaway et al. (2002) assessing links between trawling distribution and the diversity and community structure of epibenthic invertebrates and fish in the North Sea.

The broad-scale, long-term patterns in trawling distribution presented here confirmed spatial patterns described by shorter-term studies on trawling effort – such as the spatial gradient in beam trawl effort (Fig. 3), closely matching the depth gradient in the North Sea and the associated distributions of the key target species, sole *Solea solea* and plaice *Pleuronectes platessa* (e.g. Engelhard et al., 2011; van Keeken et al., 2007; Rijnsdorp et al., 1998). It is worth noting that, if analysed at a much finer spatial scale than ICES rectangles, the spatial distribution of beam trawling is much more patchy and localised, again reflecting local distributions of flatfish, and competitive interactions between fishing vessels (Rijnsdorp et al., 1998, 2000). Likewise, the distribution of otter trawling across the North Sea, when analysed at the scale of ICES rectangles (Fig. 4), appears smooth and broad. It is found to be much more patchy when analysed at finer spatial scales, as has been made

possible by the introduction of VMS (Vessel Monitoring System) on EU fishing vessels in the early 2000s (e.g. Lee et al., 2010). While VMS data provides a powerful tool for monitoring, analysing and describing fishing effort distribution, no such data is available prior to the start of the twenty-first century. By contrast, the logbook-based dataset presented here – albeit less spatially detailed than VMS data – does go back to the 1980s, allowing systematic, long-term comparisons of trawling impacts on fish, benthic invertebrates, and other organisms living on or near the seabed of the North Sea (Collie et al., 2017; Hiddink et al., 2006).

The long-term reduction in both beam and otter trawling fishing hours in the North Sea, which is evident from our reconstructed time-series, is closely associated with the European Union fleet reduction scheme, adopted since the turn of the Millennium (Villasante, 2010). This scheme, in which decommissioning of fishing vessels was paramount, was instigated specifically to address overcapacity in the European fishing fleet, and significant concerns of overfishing of key commercial fish stocks including sole, plaice, cod and sandeel (Bannister, 2004; Villasante, 2010). Since then, with the reduction in total trawling effort, strict quota regulations and the introduction on long-term management plans, several North Sea fish stocks have indeed recovered, most notably North Sea plaice (ICES, 2017). There is also evidence of recovery in the Large Fish Indicator (LFI), an OSPAR indicator of good environmental status in marine foodwebs, in response to reduced trawling pressure (Engelhard et al., 2015).

With these positive signs, it is worth noting that trawling remains one of the most pervasive anthropogenic pressures in the North Sea (Kenny et al., 2018), and it will continue to be important to monitor and assess its impacts on marine fauna and habitats. Moreover, it is very likely that the observed reduction of hours of otter trawl fishing since the 1990s would be partially –or even fully– offset in many cases by increases in vessel size, engine power, gear size and other technological developments that have taken place over these decades (e.g., see Eigaard et al., 2014). Consequently, fishing pressure and impacts on e.g. target stocks, seabed habitats or bycatch species is unlikely to have declined to the same extent that fishing hours have been reduced. Kilowatt hours of fishing may be a more useful metric to study trawling impact. However, the relevant data is not available for all countries over the time period of the study. Attempts have been made to model the impact of these technological developments on fisheries (e.g., see Eigaard et al., 2011), and could be considered for some applications of the trawling hours dataset produced in the present study.

We have previously argued that a lack of multidecadal, spatially detailed data on trawling effort has hampered attempts to study the long-term environmental footprint of trawling. The present dataset – mostly based on compiled (nominal) effort data, and for a smaller part on estimated (modelled) data – may help overcome this. For the North Sea, long-term datasets on a range of biotic and abiotic variables already exist. These include time-series of sea surface and sea bottom temperature (e.g., MacKenzie and Schiedek, 2007; Morris et al., 2018); on phytoplankton abundance and primary production (Capuzzo et al., 2018; Reid et al., 2003); on water turbidity (Capuzzo et al., 2015) and on hydrodynamics (van Leeuwen et al., 2015). Long-term data on the North Sea fish fauna, collected on International Bottom Trawl Surveys (IBTS) and Beam Trawl Surveys (BTS), is held in the 'Datras database' of ICES (e.g. Hofstede and Daan, 2008; http://www.ices.dk/marine-data/data-portals/Pages/DATRAS.aspx), and ICES also holds data on international fisheries landings dating back to the year

1903 ([https://www.ices.dk/marine-data/dataset-collections/Pages/Fish-catch-and-stock-assessment.aspx](https://www.ices.dk/marine-data/dataset-collections/Pages/Fish-catch-and-stock-assessment.aspx)). The Continuous Plankton Recorder (CPR) data provides an excellent source on zooplankton, phytoplankton, and ichthyoplankton data (Lynam et al., 2013; McQuatters-Gollop et al., 2017). These sources are now complemented by our long-term, trawling effort dataset.

Two papers, based on the present data in combination with ecological data, have already been submitted – one on 'threshold' impacts of trawling pressure on North Sea benthos (Couce et al., accepted) and one on feeding guilds within the fish community of the North Sea, in relation to fishing pressure, climate change and other drivers (Thompson et al., submitted). We encourage the use of the spatio-temporal dataset on trawling effort provided here to all those working in the fields of marine science, management and policy, who have ecosystem conservation and sustainability of marine living resources at heart, both of which are aided by a better understanding of the long-term impact from this major, widespread anthropogenic pressure.

## 6. Author contribution

E.C. conceived the research idea, E.C. and G.E. compiled the effort data and designed the methodology to reconstruct missing data and E.C. carried out the reconstruction. All authors participated in the interpretation of results and the writing of the manuscript and gave final approval for publication.

## 7. Acknowledgements

This study was supported by the Department for Environment, Food and Rural Affairs of the UK via project SLA44: Marine Biodiversity Advice, and by the Natural Environment Research Council and Department for Environment, Food and Rural Affairs via grant NE/L003279/1 (Marine Ecosystems Research Programme). We are grateful to Ole Ritzau Eigaard (National Institute of Aquatic Resources [DTU-Aqua], Denmark), Simon Jennings (International Council for the Exploration of the Sea), and Leonie Robinson (University of Liverpool) for providing datasets on North Sea trawling effort that have contributed to the reconstructed dataset. Ole Ritzau Eigaard additionally provided useful feedback regarding the effects of technological developments on trawling's ecological impacts. For the period post-2000, we also acknowledge our international STECF colleagues for compiling and quality-assuring European fishing effort data. The authors also wish to thank Oliver Williams for his help in preparing and publishing the reconstructed trawling effort data. The study benefited from feedback from Christopher Lynam and Angela Muench. The authors declare no conflict of interest.

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

**Figures**

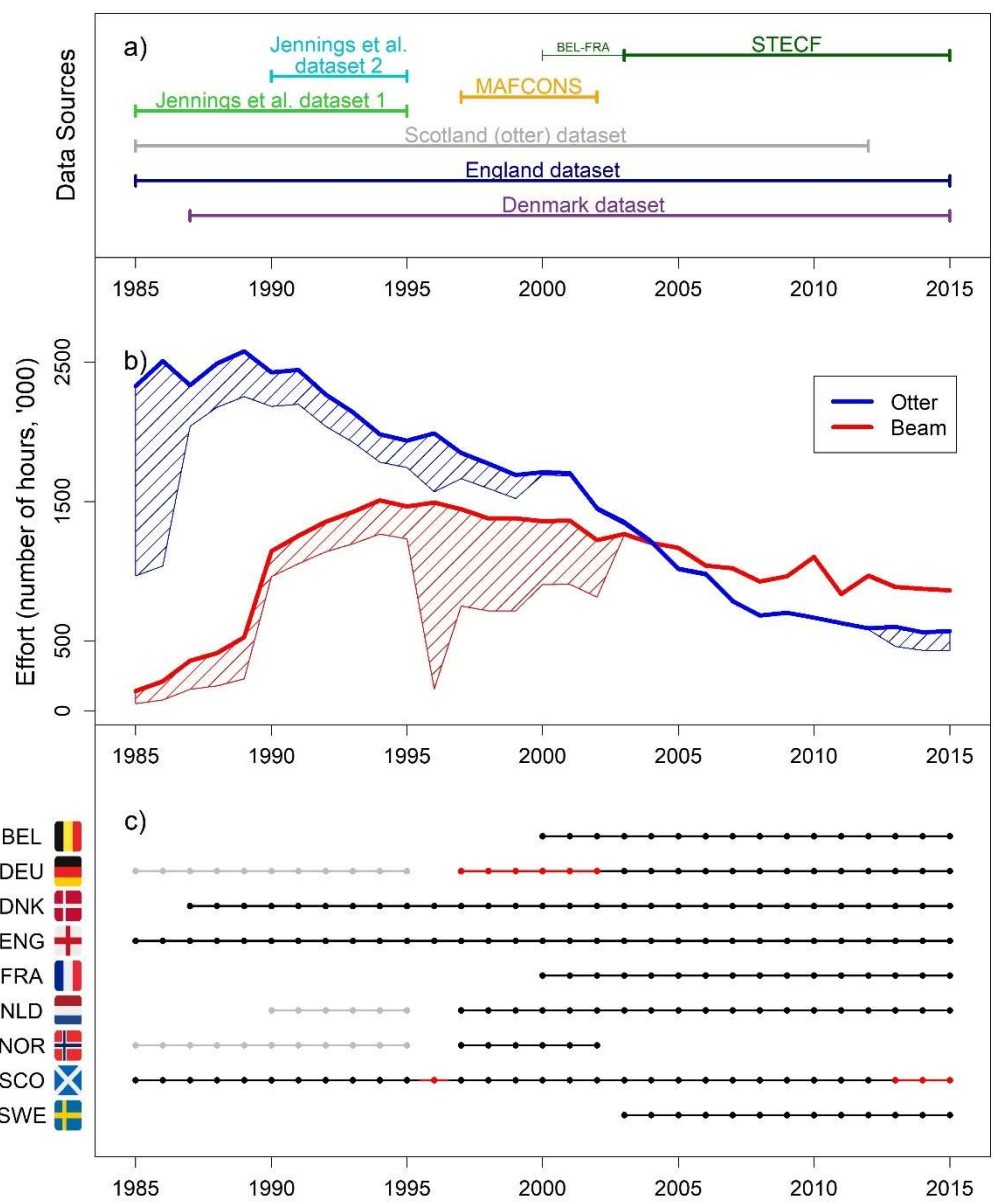

Figure 1: (a) The timelines for seven sources of compiled (nominal) fishing effort data, included in the present study; see methods section for full detail of each dataset. (b) Reconstructed total fishing hours in the North Sea by beam (red) and otter trawlers (blue), from 1985 to 2015. White-shaded areas show the proportions of the reconstructed total based on compiled (nominal) fishing effort data, and dashed areas show the proportions based on estimated (modelled) data. (c) The timelines, by country, for which nominal effort data were available, and compiled for this study. The periods shown in grey indicate years for which country data was available but only as part of a compiled set, and the individual country contribution to the total was unknown (this is data which therefore cut not be used to estimate missing periods). The periods shown in red indicate years for which only part of the data was available, or there was an issue with the compiled data.

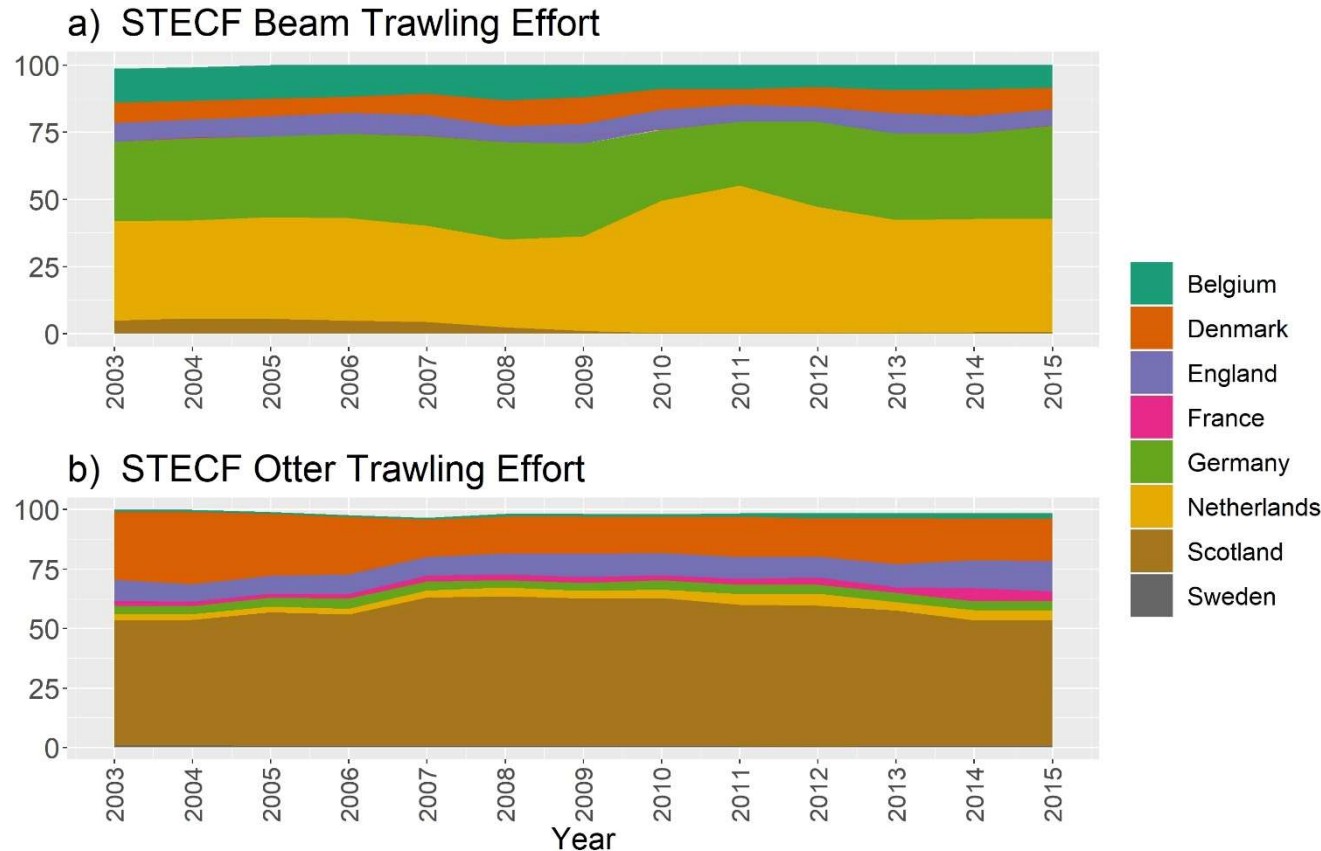

**Figure 2: Percentage contribution of individual countries over time to (a) total beam trawl effort and (b) total otter trawl effort in the North Sea, based on the STECF dataset.**

Beam trawling per year

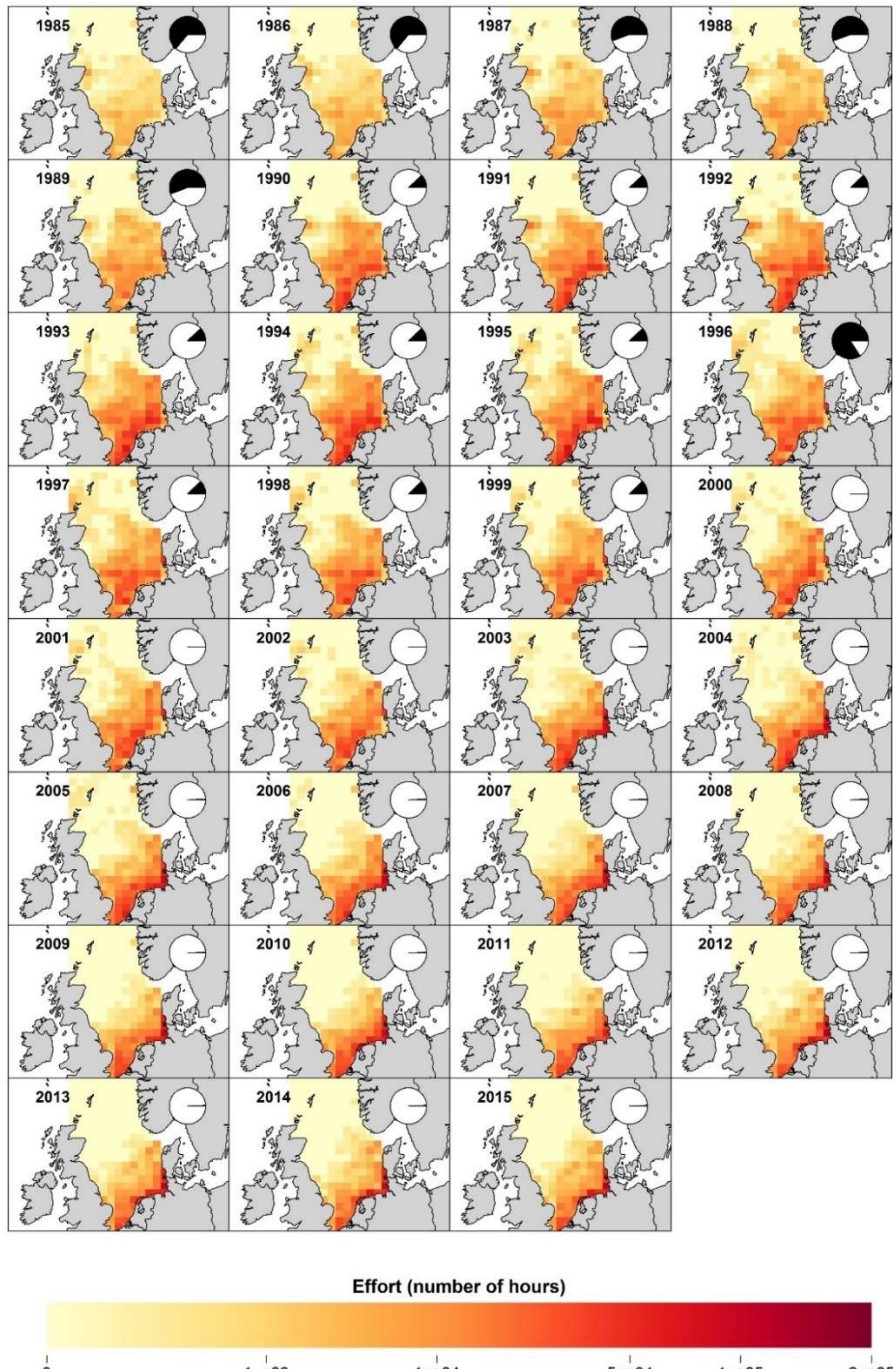

**Figure 3: Spatial distribution of beam trawling effort (number of hours trawling per ICES rectangle) in the North Sea in 1985–2015. Pie charts in the top right corners of each plot show the proportions of reconstructed trawling effort sourced from compiled (nominal) data (white) and estimated (black).**

Otter trawling per year

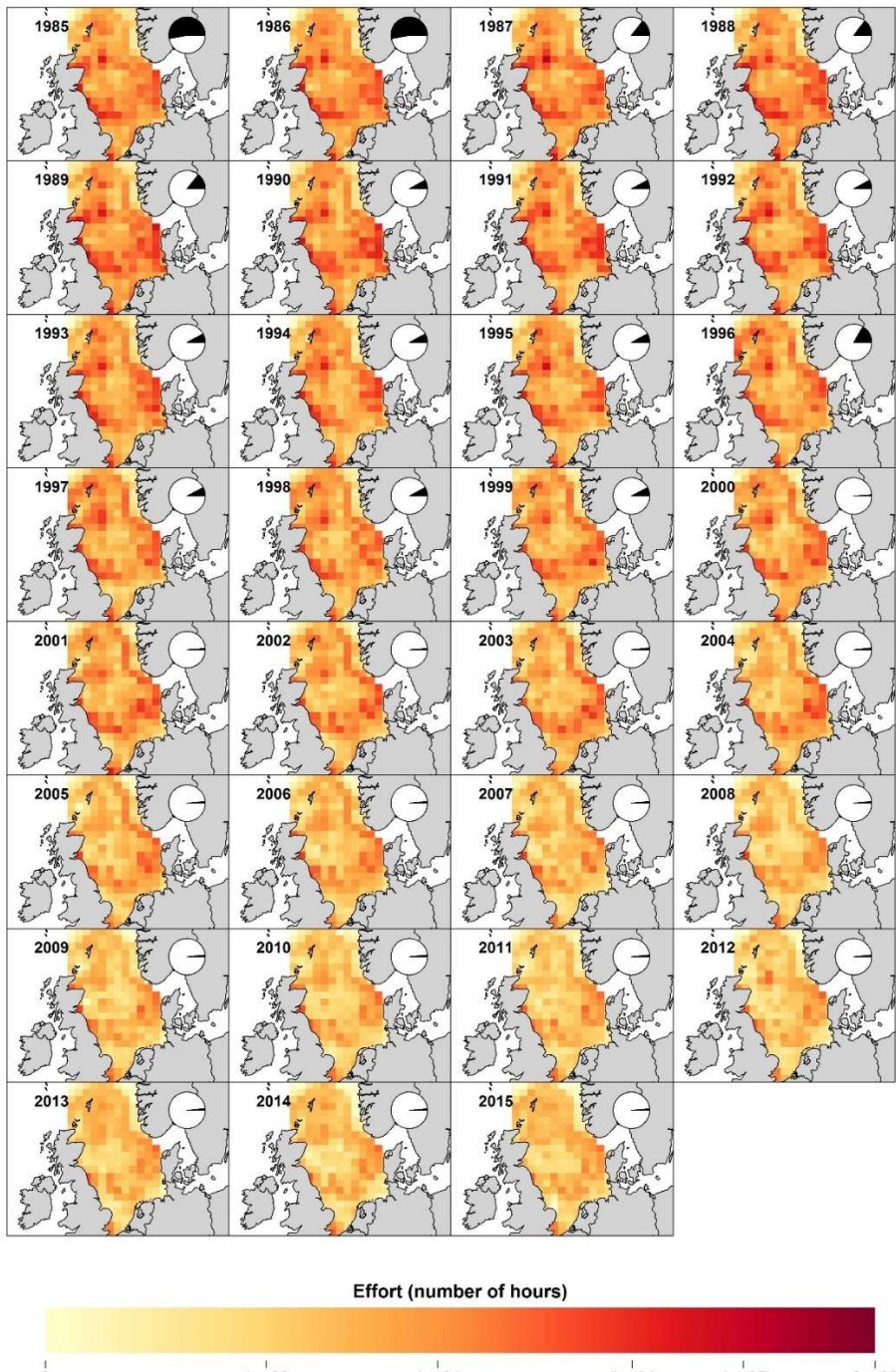

Effort (number of hours)

0   1e+03   1e+04   5e+04   1e+05   2e+05

**Figure 4: Spatial distribution of otter trawling effort (number of hours trawling per ICES rectangle) in the North Sea in 1985–2015. Pie charts in the top right corners of each plot show the proportions of reconstructed trawling effort sourced from compiled (nominal) data (white) and estimated (black).**

Relative error for reconstructed beam trawling per year

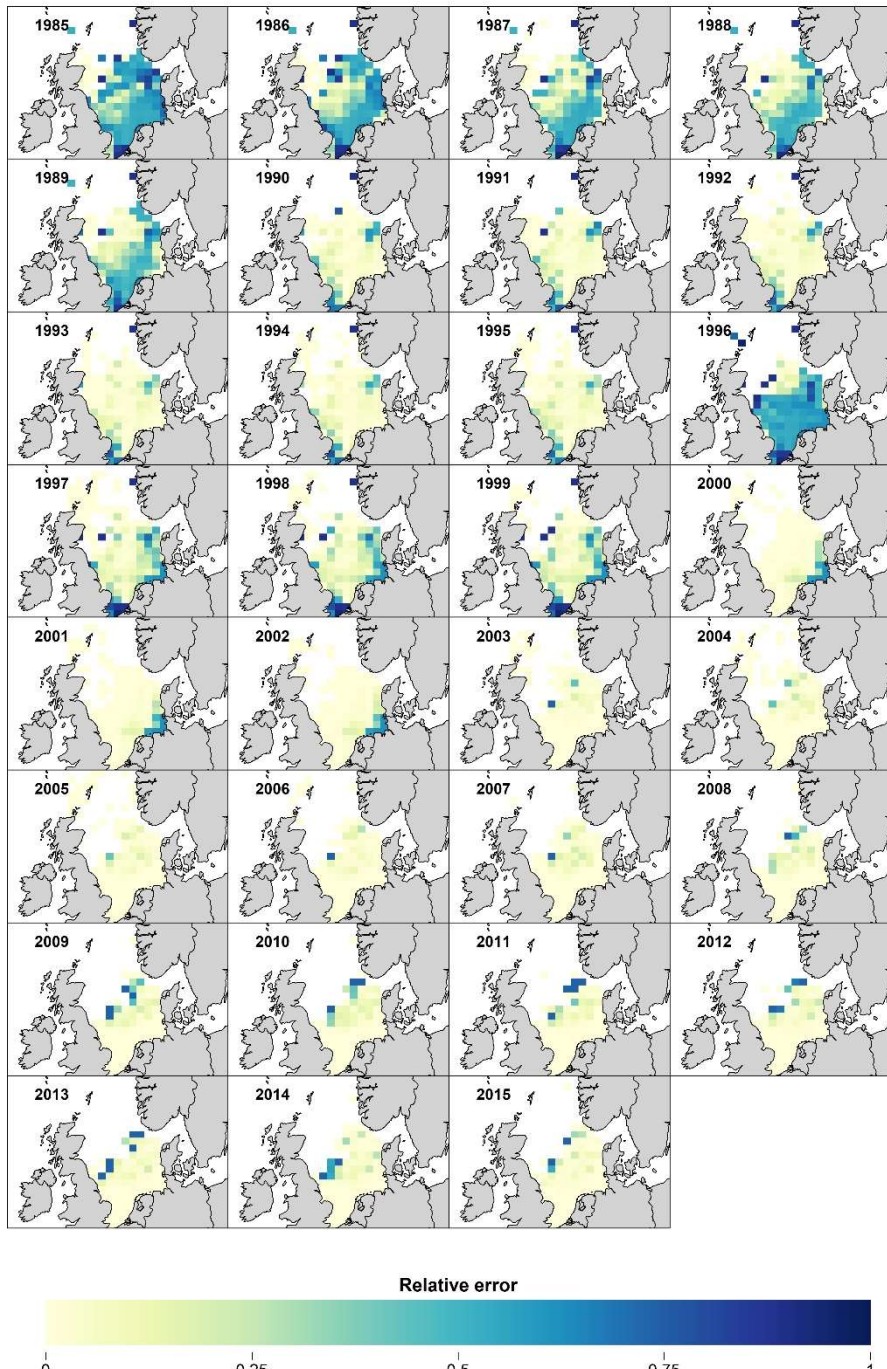

**Figure 5: Spatial distribution of relative error of the reconstructed beam trawling effort (i.e. the ratio between the error of the estimated effort and the total effort compiled + estimated) per ICES rectangle in the North Sea in 1985–2015.**

Relative error for reconstructed otter trawling per year

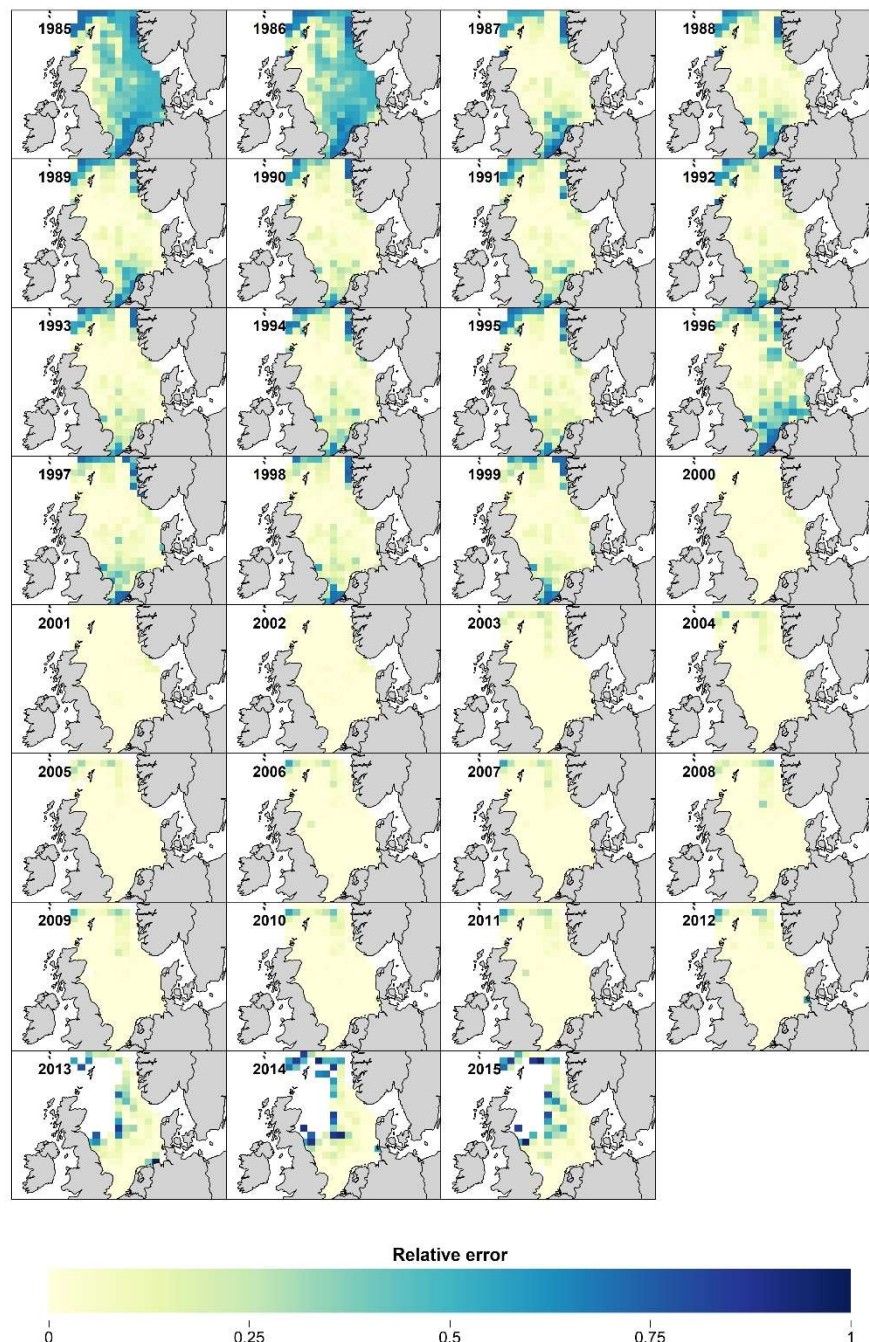

**Figure 6: Spatial distribution of relative error of the reconstructed otter trawling effort (i.e. the ratio between the error of the estimated effort and the total effort compiled + estimated) per ICES rectangle in the North Sea in 1985–2015.**

**Tables**

Table 1: Data used in the estimation of periods of missing data for each of the countries. An average spatial distribution of effort is calculated from the data in the "Source data for spatial distribution" column, and scaled to maintain the ratio of trawling by the target country with respect to the countries listed in the "Method for estimating total trawling" column.

| Country | Missing period | Gear | Source data for spatial distribution | Method for estimating total trawling |
|---|---|---|---|---|
| Germany | 1996 | Otter | 1997 MAFCONS data | • Comparing with DNK + ENG + SCO in 1997-2012. |
| Germany | 1996-2002 | Beam | 2003-2009 STECF data | • For 1996 comparing with DNK + ENG in 2003-2015.<br>• For 1997-1999 comparing with DNK + ENG + NLD + SCO in 2003-2015.<br>• For 2000-2002 comparing with DNK + ENG + NLD + SCO + BEL + FRA in 2003-2015. |
| Belgium & France | 1985-1999 | Beam, Otter | 2000-2014 STECF data | • For 1985-1986 comparing with ENG + SCO + DEU + NOR in 2000-2002.<br>• For 1987-1989 comparing with ENG + SCO + DEU + NOR + DNK in 2000-2002.<br>• For 1990-1995 beam trawling comparing with ENG + SCO + DEU + NOR + NLD + DNK, and for otter trawling comparing with ENG + SCO + DEU + NOR + NLD, in 2000-2002.<br>• For 1996 beam trawling comparing with ENG + DNK in 2000-2015, and for otter trawling comparing with ENG + DNK + SCO in 2000-2012.<br>• For 1997-1999 comparing with ENG + SCO + DNK + DEU + NLD + NOR in 2000-2002. |
| Denmark | 1985-1986 | Beam, Otter | 1987-1988 DNK data | Comparing with ENG + SCO + DEU + NOR in 1997-2002. |
| Scotland | 1996 | Beam | 1997 MAFCONS data | Comparing with ENG + DNK in 1997:2015. |
| Scotland | 2013-2015 | Otter | 2013-2015 STECF data | STECF SCO otter data scaled by the average ratio between Scotland (otter) dataset and STECF otter data in 2003-2012. |
| Norway | 1996 | Beam, Otter | 1997 MAFCONS data | For beam trawling comparing with ENG + DNK and for otter trawling with ENG + DNK + SCO in 1997-2002. |
| Norway | 2003-2015 | Beam, Otter | 1997-2002 MAFCONS data | For beam trawling comparing with ENG + DNK+ SCO + NLD and for otter trawling with ENG + DNK + SCO + NLD + DEU in 1997-2002. |
| Sweden | 1985-2002 | Otter | 2003-2015 STECF data | • For 1985-1986 comparing with ENG + SCO + DEU + NOR in 2003-2012 (using estimated NOR data).<br>• For 1987-1989 comparing with ENG + SCO + DEU + NOR + DNK in 2003-2012 (using estimated NOR data).<br>• For 1990-1995 comparing with ENG + SCO + DEU + NOR + NLD in |

| | | | | |
|---|---|---|---|---|
| | | | | 2003:2012 (using estimated NOR data).<br>• For 1996 comparing with ENG + DNK + SCO in 2003-2012.<br>• For 1997-1999 comparing with ENG + SCO + DNK + DEU + NLD in 2003-2012.<br>• For 2000-2002 comparing with ENG + SCO + DNK + DEU + NLD + BEL + FRA in 2003-2012. |
| Netherlands | 1985-1989 | Beam, Otter | 1997-2001 MAFCONS data | • For 1985-1986 comparing with ENG + SCO + DEU + NOR in 2003-2015 (for beam trawling, using reconstructed NOR data), and in 1997:2002 (for otter trawling).<br>• For 1987-1989 comparing with ENG + SCO + DEU + NOR + DNK in 2003-2015 (for beam trawling, using reconstructed NOR data), and in 1997:2002 (for otter trawling). |
| Netherlands | 1996 | Beam, Otter | 1997 MAFCONS data | For beam trawling comparing with ENG + DNK in 1997-2015, and for otter trawling with ENG + DNK + SCO in 1997-2012. |

**Table 2: Periods for which trawling effort was estimated for the quantification of errors for each of the countries, and the data used for each of the estimations. The last column shows the median error, where errors were calculated as of the absolute values of the relative differences between estimated and compiled data for all ICES rectangles and all years.**

| Country | Estimated period | Gear | Source data for spatial distribution | Method for estimating total trawling | Median error: $\frac{\mid\text{estimated} - \text{compiled}\mid}{\text{estimated}}$ |
|---|---|---|---|---|---|
| Germany | 1997 | Otter | 1998 MAFCONS data | Comparing with DNK + ENG + SCO in 2000-2012 | 0.71 |
| Germany | 2003-2008 | Beam | 2009-2014 STECF data | Comparing with DNK + ENG + NLD + SCO in 2009-2015. | 0.62 |
| Belgium & France | 2000-2007 | Beam, Otter | 2008-2015 STECF data | Comparing with DNK + ENG + NLD + BEL/FRA (BEL when estimating data from France, and vice versa) in 2008:2015. | BEL Beam: 0.89 BEL Otter: 0.75 FRA Beam: 1.82 FRA Otter: 0.70 |
| Denmark | 1987-1988 | Beam, Otter | 1989-1990 DNK data | Comparing with ENG + SCO + DEU + NOR in 1997-2002. | Beam: 0.68 Otter: 0.49 |
| Scotland | 1997 | Beam | 1998 MAFCONS data | Comparing with ENG + DNK in 1998-2015. | 1.05 |
| Scotland | 2013-2015 | Otter | 2013-2015 STECF data | Error was estimated looking at the relative differences between STECF otter data for Scotland and the Scotland (otter) dataset in 2003-2012. | 2.86 |
| Norway | 1997-1999 | Beam, Otter | 2000-2002 MAFCONS data | For beam trawling comparing with ENG + DNK+ SCO + NLD and for otter trawling with ENG + DNK + SCO + NLD + DEU in 2000-2002. | Beam: 0.74 Otter: 0.56 |
| Sweden | 2003-2008 | Otter | 2009-2015 STECF data | Comparing with ENG + SCO + DNK + DEU + NLD + BEL + FRA in 2009-2015. | 0.67 |
| Netherlands | 1997-2002 | Beam, Otter | 1997-2001 STECF data | For beam trawling comparing with ENG + DNK+ SCO in 2003-2015 and for otter trawling with ENG + DNK + SCO + NLD + DEU in 2003-2012. | Beam: 0.52 Otter: 0.70 |