# Peer review of "Reconstructing three decades of total international trawling effort in the North Sea"

_Earth System Science Data, 2019_

## Referee Comment (RC1) · Anonymous Referee #1 · 27 Sep 2019

General Comments:

The manuscript presents a dataset of trawling effort in the North Sea, comprised of compiled as well as estimated data. The authors clearly explain why there is a need for such a dataset and ways in which it can be useful to future scientific studies. The manuscript is well written, and the language is clear and easy to understand.

The link for the dataset is functional, and metadata are included on the linked page. The map tab on the linked page shows the North Sea location. The map lacks the functionality of being able to display the trawl data, but this functionality is not critical. The data files (.csv and .shp) can be easily downloaded and opened. However, when viewing the shapefile in ArcCatalog, there are no metadata associated with the file. I recommend adding metadata to the file.
The inclusion of the estimated data is what makes this dataset unique, as it otherwise would only be a compilation of datasets already in existence. It is therefore critical to have a measure of validity of these estimated data. While the authors acknowledge that there are errors associated with the estimated data, these are not quantified. The authors should consider if there are any methods which would be appropriate to validate their estimated data. For example, are there data outside of the study period (1985-2015) that could be used in order to conduct validation? If not, could data be removed and used as a testing dataset in order to statistically analyze how their methods perform? If the author thinks validation in this manner would not be appropriate or possible, they should consider whether there are any other methods by which they could quantify error. The other point that needs to be addressed with the estimated data is in regards to the methods used to select the length of the time period utilized in calculating average spatial distribution. This will be explained in the following section.

Specific Comments:

When performing the trawling effort reconstruction, the authors clearly state what data are being used to calculate the average spatial distribution of effort for each country. However, it is not explained why certain time periods are used and why different lengths of data are used for different countries. For example, for the 1985-1986 reconstruction, the Denmark spatial distribution is based on data from 1987-1989 (3 of the available 29 years for Denmark), whereas the French spatial distribution is based on data from 2000-2015 (16 of the available 16 years for France). Why was the number of years used not kept consistent when possible? How were decisions made about what length of time to use? This should either be kept consistent when possible (when the data are available to allow for it), or the authors should explain why using differing lengths of time is a more appropriate method.

If spatial distribution is assumed to change gradually over time as is stated in the assumptions, using a long time period when calculating average spatial distribution may result in loss of temporal specificity. Therefore, if a long time period is used for calculating average spatial distribution, it should be explained why this is appropriate.

Assumption 2 of the trawling effort estimates acknowledges that under particular circumstances, major changes may occur in spatial distribution. Were any major changes seen in the compiled trawling data? If so, how was this considered when estimating data?

It is not made clear why there is no beam trawling data for Sweden. Is beam trawling not occurring, or is it occurring but there are no data?

The listed countries are the most significant contributors to trawling in the North Sea, but are there other countries also trawling here? If so, approximately how much of the trawling effort can be attributed to the countries included in this study? Can an approximate quantification be given?

I found the description of the explanation for the discrepancy between the STECF data and the Scotland and England data (presented on page 9) to be not entirely clear. It is also unclear whether similar discrepancies would have been expected between STECF and other countries if data for other countries had been available. Did similar discrepancies exist for Denmark (the other country with a country-specific dataset)?

Technical Corrections:

There is currently inconsistency in indenting paragraphs in the introduction section (some paragraphs are indented at the beginning and some are not).

There is inconsistency in whether the word 'data' is used as singular or plural throughout the manuscript.

In the abstract, the following sentence has potentially confusing wording: "The dataset was largely reconstructed using compiled effort data from 7 fishing effort time-series, each covering shorter time spans and some of the countries fishing the North Sea only." This could be clarified in the following way: "The dataset was largely reconstructed using compiled effort data from 7 fishing effort time-series, each covering shorter time

spans and only some of the countries fishing the North Sea."

On Page 3 line 15, it states "For the 1977-1995 period...". This is confusing, since the remainder of the paper states 1985 as the beginning of the period included.

Typographical errors:

Page 8 line 15 - for the use of the data provided

Page 9 lines 15-16 - a factor of 2

Page 10 line 5 - rectangles

Page 11 line 13 - in relation to fishing pressure

---

## Referee Comment (RC2) · Anonymous Referee #2 · 10 Nov 2019

First of all, I would like to recognize the huge work done by the authors and the potential relevance of this contribution. However, I think that the present version of this dataset is characterized by some severe drawbacks. I identified three main problems in this paper:

1) the authors collated different blocks of data from different sources. Each of these datasets contains Nominal Effort (fishing effort in hours fishing) by area, for trawlers and beam trawlers. Unfortunately, it is largely acknowledged that all the impacts of these fishing gears are monotonically linked to the size / engine power of vessels. In the meantime, it is a fact that the structure of the EU fishing fleets changed a lot during the last decades. These two aspects, in combination, lead to the conclusion that the time series provided by the authors are not consistent. In other words, the values of

effort at the beginning of the time series could not be compared with those at the end. A potential solution for this issues is represented by the combined usage of data about effort and data about fleet structure, assuming that the spatial distribution of fishing activities is the same for different fleet segments (a huge assumption, of course);

2) Moreover, the efficiency of fishing gears changed (increased) during the last decades, as a consequence of the effort creep;

3) about the estimation of missing data: I think that the authors should at least cross-validate the fitting methods applied. This is possible (and not very complicated) if some random years, for which data are not missing, are selected and used to evaluate the goodness of estimations. Cross-validation is a very common class of techniques that can be adapted to different case studies.

---

## Author Comment (AC1) · 14 Jan 2020

The authors wish to acknowledge the significant contribution of the two anonymous reviewers.

A Supplement PDF file is attached which contains our responses to the comments by the two reviewers. In the file we reproduce the reviewer comments in black, followed by the authors' response in blue, with the changes in the manuscript indicated in each case. At the end of the file there is also a section titled "Note from the authors" in which we discuss an additional issue that has come to our attention regarding one of the datasets used in the study, and the steps we have taken to address it.

Additionally we also attach two new figures that have been added to the manuscript

showing the relative error of the reconstructed otter and beam trawling pressure data.

Please also note the supplement to this comment:
https://www.earth-syst-sci-data-discuss.net/essd-2019-90/essd-2019-90-AC1-supplement.pdf

——————————————————

[Figure]

[Figure]

Relative error for reconstructed beam trawling per year

**Fig. 1.**

[Figure]

Fig. 2.

**Supplement:**

General Comments:

The manuscript presents a dataset of trawling effort in the North Sea, comprised of compiled as well as estimated data. The authors clearly explain why there is a need for such a dataset and ways in which it can be useful to future scientific studies. The manuscript is well written, and the language is clear and easy to understand.

The link for the dataset is functional, and metadata are included on the linked page. The map tab on the linked page shows the North Sea location. The map lacks the functionality of being able to display the trawl data, but this functionality is not critical.

Actually the map is able to display the trawl data; clicking on a location will display the data linked to it.

The data files (.csv and .shp) can be easily downloaded and opened. However, when viewing the shapefile in ArcCatalog, there are no metadata associated with the file. I recommend adding metadata to the file.

The shapefile was not generated by the authors. It is automatically generated from the csv file by the Data Hub's GEOSERVER. Unfortunately, there is no functionality currently that allows us to embed metadata on the shapefile. The metadata is displayed on the website with the download links, as noticed by the reviewer.

The inclusion of the estimated data is what makes this dataset unique, as it otherwise would only be a compilation of datasets already in existence.

We would like to point out that while the compiled datasets were indeed in existence, not all of them were published or readily accessible (e.g., downloadable from the internet), so even without the estimated data we believe the compilation itself could also prove useful.

It is therefore critical to have a measure of validity of these estimated data. While the authors acknowledge that there are errors associated with the estimated data, these are not quantified. The authors should consider if there are any methods which would be appropriate to validate their estimated data. For example, are there data outside of the study period (1985-2015) that could be used in order to conduct validation? If not, could data be removed and used as a testing dataset in order to statistically analyze how their methods perform? If the author thinks validation in this manner would not be appropriate or possible, they should consider whether there are any other methods by which they could quantify error.

We have now carried out an estimation of the errors of the estimated trawling hours. This was done for each individual country by reconstructing a period for which there was country data, so that the reconstruction could be compared by the real data and an error estimated as the median of the relative differences in all cells in the grid. In each case the period estimated and the method of estimation was kept as close as possible to the real estimation. See lines 24-33, page 5 and Table 2 in the revised manuscript.

The other point that needs to be addressed with the estimated data is in regards to the methods used to select the length of the time period utilized in calculating average spatial distribution. This will be explained in the following section.

See our response in the following section.

Specific Comments:

When performing the trawling effort reconstruction, the authors clearly state what data are being used to calculate the average spatial distribution of effort for each country. However, it is not explained why certain time periods are used and why different lengths of data are used for different countries. For example, for the 1985-1986 reconstruction, the Denmark spatial distribution is based on data from 1987-1989 (3 of the available 29 years for Denmark), whereas the French spatial distribution is based on data from 2000-2015 (16 of the available 16 years for France). Why was the number of years used not kept consistent when possible? How were decisions made about what length of time to use? This should either be kept consistent when possible (when the data are available to allow for it), or the authors should explain why using differing lengths of time is a more appropriate method.

In this revision we have recomputed the estimated trawling in a more systematic and consistent way and explained the rules we followed in the reconstruction (lines 4-17, page 5). As we say in the manuscript, we expect the spatial distribution of trawling effort to change slowly over time, and therefore have based the reconstruction of the spatial distribution of a country's trawling effort over a missing period as follows: For $n$ missing years we have averaged the $n/2$ before and $n/2$ years after (or when this was impossible due to a lack of early data, on the $n$ years after). See lines 6-11, page 5 in the revised manuscript.

If spatial distribution is assumed to change gradually over time as is stated in the assumptions, using a long time period when calculating average spatial distribution may result in loss of temporal specificity. Therefore, if a long time period is used for calculating average spatial distribution, it should be explained why this is appropriate.

We have now used the average spatial distribution of a period of the same duration as the period with missing data, and as close as possible in time to it (lines 6-11, page 5). There will indeed be a loss of temporal specificity when reconstructing a long time period, but this is inevitable. We also account for this when estimating errors, since the periods that were reconstructed for the estimation of the errors were, when data allowed, of the same duration as the periods with missing data.

Assumption 2 of the trawling effort estimates acknowledges that under particular circumstances, major changes may occur in spatial distribution. Were any major changes seen in the compiled trawling data? If so, how was this considered when estimating data?

We were referring to major social changes, such as those at the beginning of World War II which brought fishing in the North Sea to a near standstill. We are not aware of any such significant abrupt change taking place during our study period. We have clarified this in lines 31-33, page 4.

It is not made clear why there is no beam trawling data for Sweden. Is beam trawling not occurring, or is it occurring but there are no data?

Indeed, beam trawl effort by Swedish vessels in the North Sea has been absent or negligible (although otter trawl effort is fairly considerable). For example, in 2003, 2006, 2009 and 2012 respectively, there were reportedly 16574, 10535, 8116 and 5333 hours fishing by Swedish bottom otter trawlers in the North Sea, but 0 hours beam trawling in any of these years (according to the STECF database). Within 2003-2012 no Swedish beam trawl effort was officially reported to STECF. We now mention this on the manuscript in lines 3-4, page 3.

The listed countries are the most significant contributors to trawling in the North Sea, but are there other countries also trawling here? If so, approximately how much of the trawling effort can be attributed to the countries included in this study? Can an approximate quantification be given?

The listed countries comprise >99% of trawling effort in the North Sea. There are some further countries that sporadically exert trawling effort in the North Sea; these include Northern Ireland (on average, 0. 5% of total EU trawling effort over the period 2003-2012), Ireland (0.0005%) and Jersey (0.0002%). It is of note that prior to the introduction of Exclusive Economic Zones (EEZs) in 1977, countries not bordering the North Sea did in some years exert significant trawling pressure in the North Sea (e.g. Poland and the USSR in the 1960s and early 1970s; see Kerby et al., 2012), but this is well before the period under study here. We have added a clarification about this in line 5, page 3.

I found the description of the explanation for the discrepancy between the STECF data and the Scotland and England data (presented on page 9) to be not entirely clear. It is also unclear whether similar discrepancies would have been expected between STECF and other countries if data for other countries had been available. Did similar discrepancies exist for Denmark (the other country with a country-specific dataset)?

We agree that these discrepancies are somewhat puzzling, and as mentioned in the manuscript (lines 8-17, page 8), we are not able to explain these fully, though we suspect they stem from the conversion factor from days at sea to hours (24 is used by STECF in the case of England and Scotland, which does not acknowledge steaming time and other non-fishing hours). We find no such discrepancies for Denmark, where the national effort data almost perfectly matches that of STECF. We now mention this in lines 17-18, page 8.

Technical Corrections:

There is currently inconsistency in indenting paragraphs in the introduction section (some paragraphs are indented at the beginning and some are not).

This has now been corrected.

There is inconsistency in whether the word 'data' is used as singular or plural throughout the manuscript.

Thanks for pointing this out. We have gone over the manuscript checking the instances "data" appears to make sure it is always used in the singular.

In the abstract, the following sentence has potentially confusing wording: "The dataset was largely reconstructed using compiled effort data from 7 fishing effort time-series, each covering shorter time spans and some of the countries fishing the North Sea only." This could be clarified in the following way: "The dataset was largely reconstructed using compiled effort data from 7 fishing effort time-series, each covering shorter time spans and only some of the countries fishing the North Sea."

Thanks for this, sentence has been changed as suggested.

On Page 3 line 15, it states "For the 1977-1995 period...". This is confusing, since the remainder of the paper states 1985 as the beginning of the period included.

We have replaced this with "For the earliest period until 1995" (line 14, page 3).

Typographical errors:

Page 8 line 15 - for the use of the data provided
Page 9 lines 15-16 - a factor of 2
Page 10 line 5 - rectangles
Page 11 line 13 - in relation to fishing pressure

All these have now been corrected.

**Anonymous Referee #2**

First of all, I would like to recognize the huge work done by the authors and the potential relevance of this contribution. However, I think that the present version of this dataset

is characterized by some severe drawbacks. I identified three main problems in this paper:

1) the authors collated different blocks of data from different sources. Each of these datasets contains Nominal Effort (fishing effort in hours fishing) by area, for trawlers and beam trawlers. Unfortunately, it is largely acknowledged that all the impacts of these fishing gears are monotonically linked to the size / engine power of vessels. In the meantime, it is a fact that the structure of the EU fishing fleets changed a lot during the last decades. These two aspects, in combination, lead to the conclusion that the time series provided by the authors are not consistent. In other words, the values of effort at the beginning of the time series could not be compared with those at the end.
A potential solution for this issues is represented by the combined usage of data about effort and data about fleet structure, assuming that the spatial distribution of fishing activities is the same for different fleet segments (a huge assumption, of course);

We fully agree that the structure of the fishing fleet and the fishing technologies have changed considerably over the 31 year period, and that therefore the potential impact on the seafloor of one hour of trawling effort in 1985 is likely to be very different than an hour in 2015. We already discuss this issue in lines 18-25, page 9. However, we have made it very clear throughout the manuscript that what we were compiling and reconstructing was **trawling effort**, quantified as hours of trawling by fishing vessels, and not "impact". This data is, by itself, relevant, and has a wide range of applications. For the particular application of analysing impact on benthic fauna it may be advisable to somehow "correct" this data to take into account technical developments (as we discuss in the manuscript in lines 22-25, page 9); for other applications it may not (i.e., a social study could just be interested in the number of hours themselves, possibly combined with other datasets regarding, for example, the numbers of people involved in the fishing industry, number of boats, etc). Data about the structure and evolving technology of the EU fishing fleet is unfortunately not readily available, certainly not at the spatio-temporal scale of this study. Some general estimations and approximations could be made, however we leave this as a subject for future work as it exceeds the aims of this manuscript.

2) Moreover, the efficiency of fishing gears changed (increased) during the last decades, as a consequence of the effort creep;

See our answer above for it also applies to this.

3) about the estimation of missing data: I think that the authors should at least crossvalidate the fitting methods applied. This is possible (and not very complicated) if some random years, for which data are not missing, are selected and used to evaluate the goodness of estimations. Cross-validation is a very common class of techniques that can be adapted to different case studies.

We fully agree that it is necessary to assess the reliability of the reconstructed missing data, and as the reviewer suggests we have done this by reconstructing non-missing data for each of the countries, and assessing how the reconstruction compares to the nominal data. However, instead of choosing random years we have attempted in each case to select periods that are as close as possible to the periods that had to be estimated due to missing data. See manuscript,

lines 24-33, page 5, together with Table 2. Additionally we have updated the dataset so that it now includes these errors.

Note from the authors:

In addition to the changes suggested by the reviewers, it has also come to our attention that the data for the effort of the beam trawling German fleet in the 1997-2002 period in the MAFCONS dataset does not include shrimp trawls. These are, however, included in the STECF dataset and also, we believe, in the Jennings et al datasets, and represent a significant contribution to the total beam trawling pressure in the North Sea. Therefore for consistency we have decided not to use the MAFCONS beam trawling data for Germany and have instead estimated it in this period (see lines 27-31, page 3 in the revised manuscript). As a result of this, our reconstructed total pressure in the North Sea for this period now looks quite different as reflected in the revised Fig. 1 (see comparison below).

Original reconstruction:

[Figure]

Updated reconstruction:

---

## Author Response (AR2)

**Topical Editor Decision**: Publish subject to minor revisions (review by editor) (17 Jan 2020) by Giuseppe M.R. Manzella

**Comments to the Author:**

The paper is well written and the methodology well explained.

My only suggestion is: move 'data availability' paragraph after 'discussion'. A reader would read discussion on data, results and opinion of the authors without interruptions. The 'data availability' paragraph is interrupting the authors' discourse.

Thanks a lot for your kind words. Per your suggestion we have uploaded a revised version of the manuscript with the "Data availability" section moved to the end, after the "Discussion" section. We agree it reads better this way.

Additionally we have carried out a few minor changes:
- When we mention the dataset we clarify we are referring to "Version 2" (which is the version produced after the review, including errors): line 18, page 1, and line 14, page 10.
- We now mention in the Data availability section that the "estimated error" is also included in the dataset: line 13, page 10
- We have updated the status of a reference from "accepted" to "in press": line 6, page 10.
- We have updated the "Acknowledgements" section to also mention an additional person that helped get the revised version of the data online (line 14, page 11) and the feedback received from the anonymous reviewers (line 15, page 11).

[revised manuscript text omitted]